# Occurrence and nature of questionable research practices in the reporting of messages and conclusions in international scientific Health Services Research publications: a structured assessment of publications authored by researchers in the Netherlands

Reinie G Gerrits,[ID] Tessa Jansen, Joko Mulyanto,[ID] Michael J van den Berg, Niek S Klazinga, Dionne S Kringos

Public Health, Amsterdam UMC, University of Amsterdam, Amsterdam Public Health Research Institute, Amsterdam, The Netherlands

**Correspondence to**
Reinie G Gerrits;
r.g.gerrits@amc.uva.nl

## ABSTRACT

**Objectives** Explore the occurrence and nature of questionable research practices (QRPs) in the reporting of messages and conclusions in international scientific Health Services Research (HSR) publications authored by researchers from HSR institutions in the Netherlands.

**Design** In a joint effort to assure the overall quality of HSR publications in the Netherlands, 13 HSR institutions in the Netherlands participated in this study. Together with these institutions, we constructed and validated an assessment instrument covering 35 possible QRPs in the reporting of messages and conclusions. Two reviewers independently assessed a random sample of 116 HSR articles authored by researchers from these institutions published in international peer-reviewed scientific journals in 2016.

**Setting** Netherlands, 2016.

**Sample** 116 international peer-reviewed HSR publications.

**Main outcome measures** Median number of QRPs per publication, the percentage of publications with observed QRP frequencies, occurrence of specific QRPs and difference in total number of QRPs by methodological approach, type of research and study design.

**Results** We identified a median of six QRPs per publication out of 35 possible QRPs. QRPs occurred most frequently in the reporting of implications for practice, recommendations for practice, contradictory evidence, study limitations and conclusions based on the results and in the context of the literature. We identified no differences in total number of QRPs in papers based on different methodological approach, type of research or study design.

**Conclusions** Given the applied nature of HSR, both the severity of the identified QRPs, and the recommendations for policy and practice in HSR publications warrant discussion. We recommend that the HSR field further define and establish its own scientific norms in publication practices to improve scientific reporting and strengthen the impact of HSR. The results of our study can serve as

## Strengths and limitations of the study

► Given the explorative nature of this study, we applied a broad and sensitive definition of 'questionable research practices' (QRPs) that allows for the identification of QRPs previously overlooked in related assessments.

► This study describes an assessment of publications and is therefore able to detect QRPs that go unnoticed in survey studies that rely on self-report.

► Although we aimed to develop a reliable measurement instrument that would guide the review process, the instrument allowed latitude for the reviewer's interpretation.

► In our assessment method, we relied on consensus among assessors, which inevitably introduces some subjectivity.

► Because publications were selected based on the title, selection bias might have occurred.

an empirical basis for continuous critical reflection on the reporting of messages and conclusions.

## INTRODUCTION

In 2009, Chalmers and Glasziou estimated that 85% of research funding in biomedical sciences was wasted avoidably,[1] resulting in *The Lancet*'s series 'Increasing value: reducing waste'. This series has stirred the international scientific community, prompting funders, regulators, academic institutions and scientific publishers to act. Funders of biomedical research have responded by organising conferences on research waste, and journal editors have initiated discussions on data

sharing and open access.[2] While evidence for questionable research practices (QRPs) in biomedical sciences is mounting,[1] little is known about the occurrence and nature of QRPs in the policy-oriented and management-oriented field of health services research (HSR). In particular, QRPs in the reporting of messages and conclusions have flown under the radar. The term 'questionable research practices' is commonly used to describe practices such as selective publication of results, concealing of conflicts of interests and describing a hypothesis after finding significant results.[3] A questionable practice is not necessarily wrongful but does 'raise questions'. In this study, we further define the meaning of QRPs in the reporting of messages and conclusions in the field of HSR specifically.

The HSR field is an applied field of research, and produces evidence on topics such as copayments, evaluation of quality improvement efforts, cost-effectiveness of medications, patient empowerment, therapy compliance and effects of policies. Given the growing evidence for the prevalence of QRPs in the reporting of messages and conclusions in the biomedical field,[4 5] QRPs may also occur in the HSR field. In the biomedical field, a systematic review by Chiu *et al* shows that estimates for the occurrence of questionable research practices in the interpretation of results in scientific publications vary from 10% of publications deriving discordant conclusions from study results to 100% of publications containing rhetorical practices resulting in spin, such as failure to compare risk with benefits in randomised controlled trials (RCTs).[4]

Just like biomedical researchers, health services researchers are under pressure to publish in high-impact journals to increase their citation scores and attract media attention to augment their prestige and chances for future research funding and job security.[6–9] Unlike biomedical research, HSR findings are not easily generalised from one local or national health services setting to another, and messages and conclusions tend to be limited to a specific national context.[10] A broad spectrum of quantitative and qualitative methods is used in HSR, including designs that are less subject to strict codes of execution than RCTs, such as observational and case study designs. Furthermore, HSR has difficulty creating alignment between the construction of scientific knowledge and the implementation of that knowledge in policy and practice.[11] This combination of HSR-specific characteristics may result in a different set of QRPs in the reporting of a scientific study. The variation of designs other than RCTs, as is more common in the biomedical field, might invite unjustified claims of causality. Moreover, the context-specific research may increase unjustified claims of generalisability, and the difficulty in translating knowledge to practice may result in unsupported recommendations or implications.

Although reporting in scientific publications is highly standardised, the discussion and conclusion sections offer researchers relative freedom when deriving messages and conclusions from study results.[5] We explored the occurrence and nature of QRPs in the reporting of messages and conclusions in international scientific HSR publications authored by researchers from HSR institutions in the Netherlands. We also examined the relationship between study type, methodology and design and the occurrence of QRPs. With our study, we want to fuel the debate on fostering responsible messages and conclusions, and provide a basis for the discussion on QRPs in the international HSR field.

## METHODS

### Setting
This study assessed scientific publications authored by researchers from 13 HSR groups, departments or institutions (hereafter referred to as 'HSR institutions') in the Netherlands, including both academic and non-academic institutions. These institutions all agreed to participate in an effort to assure the overall quality of HSR publications in the Netherlands.

### Defining QRPs in the reporting of messages and conclusions in HSR
We conducted a literature review on QRPs in the reporting of messages and conclusions in biomedical research and HSR.[12–14] An initial definition of QRPs in the reporting of messages and conclusions in HSR was proposed and discussed at a consensus meeting with the directors/leaders of the 13 participating institutions. This was then validated through inputs from five leading international health services researchers (10 were invited; 50% non-response), and resulted in the following amended definition: ' To report, either intentionally or unintentionally, conclusions or messages that may lead to incorrect inferences and do not accurately reflect the objectives, the methodology or the results of the study. '

### Measurement instrument
We developed an extensive list of QRPs in the reporting of messages and conclusions. Items were based on the EQUATOR checklists[15] and earlier checklists for identifying 'spin' (ie, 'a way to distort science reporting without actually lying')[5] or other QRPs.[13 14 16 17] The proposed list of QRPs was reviewed, refined and complemented using 14 semistructured interviews with the directors/leaders and representatives (n=19) of the 13 participating HSR institutions. Next, the five participating international health services researchers provided email feedback on the list resulting from these interviews; the list was adapted accordingly, resulting in 35 possible QRPs in the reporting of messages and conclusions in HSR publications.

We developed a data extraction form in Excel that contained the list of QRPs and bibliometric information, and conducted a pilot to evaluate its feasibility and usability. In the pilot, two assessors (RGG, TJ) independently assessed five international HSR publications

to identify modifications needed to improve the form, and to align the interpretation of the items. The project group discussed the proposed modifications, resulting in the final version. The data extraction form (see online supplementary material 1) and a methodology of the development of the data extraction form (see online supplementary material 2) are provided in the supplementary material.

## Sample

We aimed to include 10 HSR publications from each participating HSR institution. Inclusion criteria were: published in 2016 in an international peer-reviewed scientific journal, written in English, reporting HSR findings and first-authored and/or last-authored by researchers affiliated with the respective HSR institution. As both the first author and the research institution are likely important factors influencing the occurrence of QRPs, only unique first authors were included in the publication. Moreover, not more than 10 publications per institution were included. This will ensure a maximum spread of authors and institutions across the sample.

Publication lists of the HSR institutions were retrieved either by searching publicly accessible online sources (eg, annual reports, open repositories or the research groups' website) or obtained from secretaries or librarians. All lists were verified by the respective HSR institutions. These lists included both HSR and non-HSR publications.

Two researchers (RGG, TJ) selected all titles from the 13 publication lists that were likely to indicate empirical or systematic assessment studies in HSR. Publications were included if their title fitted the definitions of HSR by Juttmann and Lohr and Steinwachs.[18 19] These definitions are commonly used by HSR institutions (eg, in education) in the Netherlands. To select HSR studies, TJ and RGG first individually selected titles from the publication lists. Next, RGG and TJ compared their selections of titles and noted any differences. After completing the selection of the first HSR publications, selection was reviewed and approved by the research group (NSK, DSK, MJB). TJ and RGG then continued applying the selection method to the remaining publication lists. In a consensus meeting between TJ and RGG, differences in selected titles were resolved by discussing its fit with the definition. Consensus was reached on all included publications.

The HSR publications (n=717) were assigned a random number. Per institution, the publications with unique first authors with the lowest assigned number were included in the sample. Three HSR institutions did not have enough publications with unique first authors, resulting in a selection of nine, eight and two publications for these institutions. Furthermore, two publications were excluded during assessment because they concerned research protocols. These publications were replaced by another publication authored by the same institution. One publication was excluded because its methodology was considered incomprehensible by the reviewers. Ultimately, 116 HSR publications were included (16% of tot sample).

## Assessment process

Two reviewers independently assessed all publications (RGG and TJ or RGG and JM). RGG has primarily qualitative HSR experience and is trained in health economics. TJ and JM have primarily quantitative HSR experience and are trained in public health, management, economics and law and medicine, respectively.

The assessment started with a test phase. During this phase, agreements and disagreements in assessments of the first 30 publications were thoroughly discussed (by RGG, TJ, NSK and DSK) to increase the accuracy of the assessments; agreement between the two reviewers (TJ, RGG) was 81% for the first 20 publications, which increased to 82% when assessing the next 10 publications. The notion emerged that it was necessary having two reviewers with complementary expertise assess each publication independently, followed by a consensus procedure and random check by the project leaders. RGG trained the third reviewer (JM).

RGG assessed all included publications, while TJ assessed the first 59 publications, and JM the remaining 57. All data were entered in the data extraction form. QRPs were coded as either 1, 'present'; 0, 'not present'; −8, 'not applicable to this study' (primarily used for items not applicable for qualitative research); or −9, 'not assessable'. To justify their assessments, the reviewers recorded their motivation for every identified QRP. At a later stage, QRPs in implications and recommendations for policy and practice were further refined into 'not mentioned' if no implication or recommendation was included in the publications, and 'not sufficiently justified', if the authors did not provide any explanation for their implications or recommendations. The reviewers held regular consensus meetings (after review of 10 publications) to discuss and reach agreement on all identified QRPs.

During the consensus meetings, the reviewers compared their assessment of all items. Inconsistencies between the individually assessed QRPs were identified, discussed and adapted. Any remaining disagreements (n=2) were resolved by a senior researcher (DSK). NSK and DSK each reassessed a random sample of six publications, so 10% of all included publications (n=12). As a result, two identified QRPs were retracted, and two QRPs were added to the reassessed publications.

## Analysis

The characteristics of the included publications were described by calculating their occurrence with the percentage or mean number of publications.

We counted the total number of QRPs per publication, and the percentage of HSR publications with number of observed QRPs. The latter was visualised in a histogram. Occurrence of specific QRPs was calculated as a percentage of publications containing this particular QRP. The percentage of publications containing QRPs that were not applicable to qualitative research was calculated only for quantitative and mixed methods-based publications (n=83), (eg, the QRP: 'The relevance of

statistically significant results with small effect size is overstated' is only applicable to quantitative research).

We used a Kruskal-Wallis test to calculate the difference in total number of QRPs applicable to all research designs by methodological approach (quantitative, qualitative, and mixed) type of research (descriptive, exploratory, hypothesis testing and measurement instruments) and study design (observational, (quasi) experimental, systematic review, economic evaluation, case study and meta-analyses). We used the Strengthening the Reporting of Observational Studies in Epidemiology checklist for observational studies in the reporting of this research.[20] Analyses were conducted using SPSS V.24.[21]

### Patient and public involvement

No patients were involved in this study. This study was designed with the input provided by the participating HSR institutions at a consensus meeting at the onset of the study, and individual interviews with the directors/leaders of the 13 participating institutions. During a progress meeting with the participating institutions, preliminary (aggregated level) results were discussed to validate and complement the interpretation of findings.

### Ethics approval

A waiver for ethical approval was obtained for this study. To avoid negative consequences for the authors of the included publications, each publication was assigned a unique identification number. Extracted data were entered in SPSS using this number to separate author information from the study data.

## RESULTS
### Characteristics of included publications

Table 1 presents the characteristics of the 116 included publications from the 13 participating HSR institutions. To summarise, 54.3% of the publications were quantitative, 28.4% were qualitative and 17.2% applied a mixed methods approach. Sixteen per cent of the publications were based on a published study protocol. The mean impact factor of the journals was 2.81, and the average number of authors was six.

Of the 116 HSR publications, the median number of QRPs per publication was six (IQR, 5.75), out of 35 possible QRPs. The distribution of the observed frequency of QRPs across publications is visualised in figure 1.

### Frequency of QRPs per type

For each of the QRPs, we counted how often they were identified in the included publications. Online supplementary material 3 presents the percentage of occurrence per QRP type.

QRPs that occurred most frequently were:

► Implications for policy and practice do not adequately reflect the results in the context of the referenced literature (69.0%)*;

| Table 1 Characteristics of included publications | |
|---|---|
| **Total (n=116)** | **n (%)** |
| HSR domain | |
| Policy | 19 (16.4) |
| Social factors | 11 (9.5) |
| Financing systems | 10 (8.6) |
| Organisational structures and processes | 43 (37.1) |
| Health technologies | 11 (9.5) |
| Personal behaviours | 22 (19.0) |
| Methodological approach | |
| Quantitative | 63 (54.3) |
| Qualitative | 33 (28.4) |
| Mixed methods | 20 (17.2) |
| Type of research | |
| Descriptive | 31 (26.7) |
| Exploratory | 59 (50.9) |
| Hypothesis testing | 19 (16.4) |
| Measurement instruments | 5 (4.3) |
| Other | 2 (1.7) |
| Design | |
| Observational | 59 (50.9) |
| (Quasi) experimental | 9 (7.8) |
| Systematic review | 17 (14.7) |
| Economic evaluation | 5 (4.3) |
| Meta analyses | 3 (2.6) |
| Case study | 22 (19.0) |
| Other | 1 (0.9) |
| Protocol published | 19 (16.4) |
| Funder of study stated | 98 (84.5) |
| Contributions stated | 57 (49.1) |
| Number of included journals | 80 (100.0) |
| | Mean |
| Impact factor journal (n=93 publications*) | 2.81 (SD 1.45) |
| Number of authors (n=116) | 6.12 (SD 5.53) |

Occurrence of QRPs per publication.
*Not all journals had an impact factor. Mean impact factor was calculated over 93 publications.
HSR, health services research.

– *In 50.0% of publications, no implications for policy and practice were mentioned, and in 19.0% of publications, implications were mentioned without adequate justification.
► Recommendations for policy and practice do not adequately reflect the results in the context of the referenced literature (65.5%)**;
– **In 34.5% of publications, no recommendations for policy and practice were reported, and in 31.0% of publications, recommendations were mentioned without adequate justification.

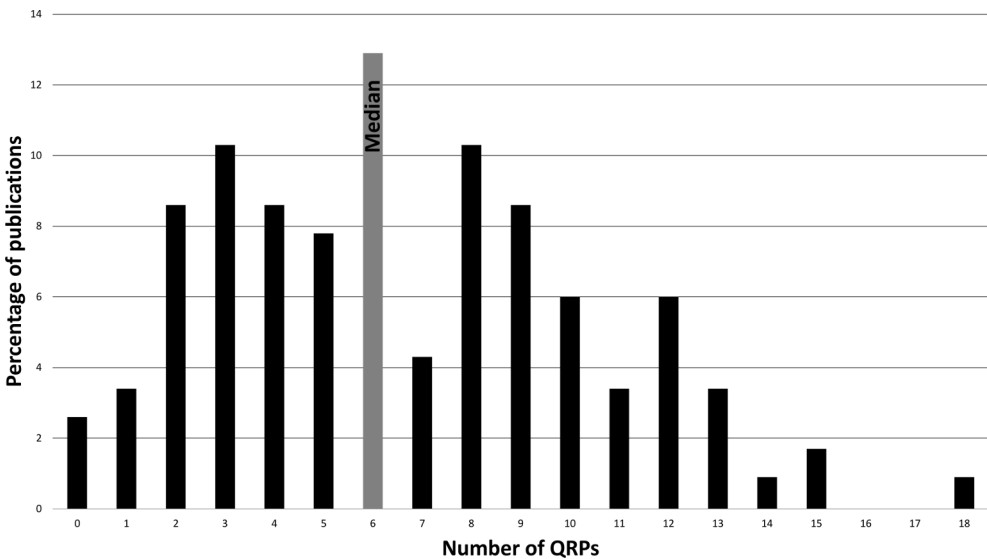

**Figure 1** Percentage of health services research publications with number of observed questionable research practices (QRPs) in the reporting of messages and conclusions.

► Contradicting evidence is poorly documented (63.8%);
► Conclusions do not adequately reflect the findings as presented in the results section (46.6%);
► Possible impact of the limitations on the results is not or poorly discussed (44.0%);
► Conclusions are not supported by the results as presented in the context of the referenced literature (43.1%).
QRPs that occurred least frequently were:
► The main source of evidence for supporting the results is based on the same underlying data (2.6%);
► Generalising findings to populations not included in the original sample is not justified (2.6%);
► Causative wording is used in the hypothesis/research question, although there is no theory to support causation (2.4%);
► Possible clinical relevance of statistically non-significant results is not addressed (2.4%);
► Generalising findings to time periods not included in the original study is not justified (0.0%).

### Distribution of QRPs
Figure 2 shows the distribution of QRPs across publications. The horizontal axis shows the publications (n=116) ordered from the publication with the lowest (0) to the highest number (18) of observed QRPs in the reporting of messages and conclusions. The vertical axis shows the QRPs ordered from least (generalisation to different time period) to most (implications for practice are lacking) frequently observed. On the right vertical axis, the occurrence of QRPs is presented in number of QRPs counted. Each dot represents a QRP.

### The difference in the number of QRPs by publication characteristics
Table 2 shows the associations between total number of QRPs (applicable to all study designs) and methodological

approach (quantitative, qualitative and mixed), type of research (descriptive, exploratory, hypothesis testing and measurement instruments) and study design (observational, [quasi] experimental, systematic review, economic evaluation, case study and meta-analyses). No statistically significant differences in number of QRPs was found by type of research, methodological approach or study design.

### DISCUSSION
We explored the occurrence and nature of QRPs in the reporting of messages and conclusions in international scientific HSR publications authored by researchers from HSR institutions in the Netherlands, and examined the relationship between study type, methodology and design and the occurrence of QRPs. Our results indicate that HSR publications have a median of six QRPs per publication. We identified most QRPs in the reporting of implications for policy and practice, recommendations for policy and practice, contradictory evidence, study limitations and conclusions based on the results and in the context of the literature. No significant associations between number of QRPs and type of study, study design or methodological approach were identified.

### Limitations and strengths
We applied a broad and sensitive definition of 'questionable', for instance by considering the absence of contradictory evidence or the absence of implications and recommendations for policy and practice as a QRP. The choice to not present contradictory evidence does not defy current publication checklists, yet this practice may hinder interpretation of findings in the full context of evidence. If authors searched for contradictory evidence, but did not mention its absence, readers of the publication would not have any clues on its existence.

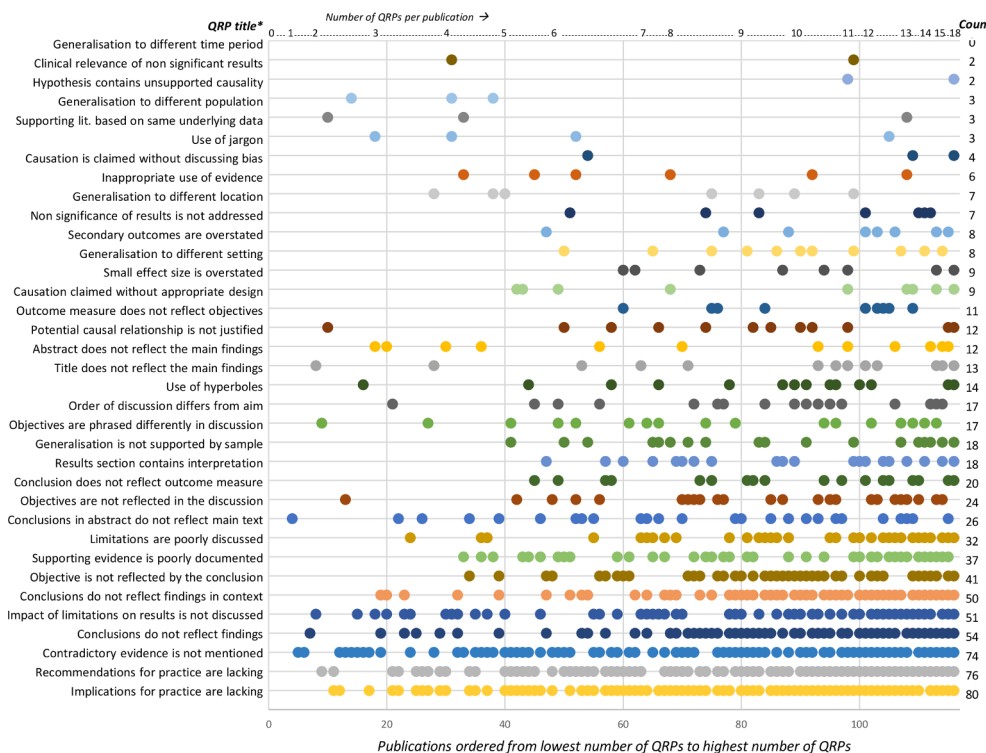

**Figure 2** Distribution of questionable research practices (QRPs) in the reporting of messages and conclusions across health services research publications, ordered from lowest to highest number of observed QRPs.

*The full QRP name is provided in supplementary material 3, table 1 ordered from least frequently found (Generalisation to different time period) to most frequently found (Implications for practice are lacking) QRP.*

Knowledge on the occurrence of QRPs is often derived from survey studies, relying on self-report.[3] These studies focus on the knowledge of consciously conducted, well-known QRPs. Our assessment approach allowed us to gain insight in less severe, more likely unconsciously occurring QRPs in the reporting of messages and conclusions specifically. The number of QRPs identified through assessment is generally higher than in studies relying on self-report.[3 4] With our broad definition encompassing 35 possible QRPs, we bring to light the areas that offer possibilities for further enhancing publication practices in HSR. Consequently, this definition allows for a discussion in the field of HSR on the extent to which the identified QRPs are acceptable. This is an important strength of our applied approach.

Although we endeavoured to develop a reliable measurement instrument that would guide the review process, the instrument allowed latitude for the reviewer's interpretation. Consequently, a different group of reviewers might arrive at somewhat different scoring frequencies for observed QRPs. However, because we defined each QRP in detail, it is unlikely that there would be substantial differences in the overall distribution of different types of QRPs across publications. Our consensus method contains a degree of subjectivity, and there is the risk that one reviewer's opinion will dominate. To counteract this, NSK and DSK performed random checks on 10% of all assessments. By recording the motivation for every identified QRP, we supported the consistency of our measurement and justified our results. Because publications were selected based on the title, selection bias might have occurred. Considering we

**Table 2** Association between total number of questionable research practices and type of research, methodological approach and study design

|  | Median | 95% CI | P value |
|---|---|---|---|
| **Methodological approach** |  |  | 0.339 |
| Quantitative | 5 | 4.88 to 6.43 |  |
| Qualitative | 6 | 4.98 to 7.62 |  |
| Mixed methods | 7 | 5.34 to 8.46 |  |
| **Type of research** |  |  | 0.295 |
| Descriptive | 6 | 4.77 to 6.78 |  |
| Exploratory | 7 | 5.76 to 7.60 |  |
| Hypothesis testing | 4 | 3.40 to 6.81 |  |
| Measurement instruments | 5 | 2.14 to 6.66 |  |
| Other | 5 | −33.12 to 43.12 |  |
| **Study design** |  |  | 0.159 |
| Observational | 6 | 5.56 to 7.21 |  |
| (Quasi) experimental | 3 | 2.07 to 5.71 |  |
| Systematic review | 6 | 4.61 to 8.33 |  |
| Economic evaluation | 4 | 1.61 to 7.59 |  |
| Case studies | 6 | 4.71 to 8.01 |  |
| Meta-analyses | 5 | 0.50 to 10.84 |  |

found no relationship between study characteristics and number of QRPs, it is unlikely that a different sample would have led to different results. Inevitably, reviewers sometimes assessed publications written by authors they knew professionally or personally, and as such, a positive view of a colleague's work might have led to underestimating the QRPs in these publications.

Our study results may be representative for HSR research publications internationally. Given the fact that publication in international journals is highly standardised in terms of language (English) and format, our findings can most likely be transferred to HSR communities in other countries.

## Interpretation

In HSR publications, recommendations for policy and practice warrant most attention. A study can be conducted properly, using a sound design and appropriate methodology. However, making recommendations without adequate justification could lead to incorrect inferences in policy and the management of healthcare, and undermine society's confidence in science.[11 22–25]

Measures for safeguarding scientific soundness like those often used in biomedical research (eg, trial registration, open data policies and an improved reporting and archiving infrastructure[26]) do not address reporting conclusions not supported by study results, and are not tailored to the observational and explorative designs most prevalent in HSR. Moreover, existing publication checklists address a report's completeness, but do not question the justification of the conclusions.[5] If we intend to improve the reporting of HSR conclusions and recommendations, we will need to better understand the factors that influence authors when reporting the discussion and conclusions section of an HSR publication, for example, media pressure and relationships with funders.[6 7 9 27] Journals may have influence on the reporting of a study through control of the review process.[28] Moreover, research institutions may prevent the occurrence of QRPs by enhancing internal integrity, training in scientific writing and communication among researchers.[29] Consequently, subsequent research can focus on what influences researchers when writing their scientific publications, and what factors play a role in the process from research design to the acceptance of a manuscript by a peer-reviewed journal.

A third of the HSR publications studied gave no recommendations for policy or practice, while another third did not provide an adequate justification for the recommendations. One could argue that HSR is an applied field of research, and that its ultimate goal should be to contribute to better health services and systems; researchers should therefore take responsibility for providing guidance to those who can act on the research findings instead of leaving them empty-handed. On the other hand, health services researchers may feel more comfortable committing to a more traditional interpretation of the role of academics, refraining from normative judgement. If

the latter is the dominant viewpoint, the HSR community needs to consider the role of scientific evidence in helping decision makers address the challenges they face, and informing policies and practices. Internationally, the HSR community has been promoting further strengthening of the link between HSR and practice.[30]

In biomedical research, research being 'new' might contribute to a confused assessment of implications.[31] This problem is amplified in HSR, where there is a limited accumulation of evidence. HSR considers a larger range of contextual factors and stakeholders in politics or management. Moreover, HSR recommendations are often based on observational or exploratory research, which is considered to be weak evidence in biomedical circles (eg, the Grading of Recommendations, Assessment, Development and Evaluation (GRADE) checklist).[32] Perhaps the norms determined by the biomedical research field make health services researchers hesitant to provide any implications or recommendations at all.

## Implications and recommendations for policy and practice

The HSR field currently seems to adhere to the norms and expectations set by the biomedical field, even though HSR is multidisciplinary, and differences in approach and type of methodology pose serious challenges to observing these norms. Therefore, the HSR community needs to further define specific scientific norms appropriate to the field.

Scientific norms are developed through the forum of a scientific community.[33] This forum function is particularly strong in the Netherlands, where a community of HSR institutions work together closely. Our study was able to bring together the main Dutch academic and non-academic HSR institutions. Consequently, the results of our study help to facilitate critical reflection on the current state of research and encourage debate on how to systematically advance the reporting of messages and conclusions in HSR. Such a debate in the Dutch context is needed, given the attempts over the past decade by the Netherlands Organisation for Health Research and Development (ZonMw) to strengthen the link between research and practice. It would also be very timely, considering the ongoing, overarching Dutch research programme on responsible research practices funded by ZonMw, of which this study is a part. We recommend the HSR community to reflect on the questions our results bring forward: how do we include implications and recommendations for policy and practice in scientific publications?; how should we describe conclusions in context of literature with limited accumulation of evidence?; and what is the severity of the identified QRPs? Through this publication, we would like to urge journal editors and those working in the international field of HSR to join in this debate. After establishing norms regarding these frequently occurring QRPs, journal editors and HSR institutions may contribute to the prevention of QRPs by implementing strategies tailored to HSR research specifically.

## CONCLUSIONS

QRPs in the reporting of messages and conclusions occur frequently in peer-reviewed international scientific HSR publications from Dutch HSR institutions. These QRPs differ in severity and cannot always be qualified as wrongful, but they do 'raise questions'. To ensure the applicability of HSR research in policy and practice, the HSR field should reflect on scientific norms for the reporting of conclusions and the inclusion of recommendations for policy and practice. Our study can serve as an empirical basis for continuous critical reflection on the current state of research, and encourage debate on how to systematically advance the reporting of messages and conclusions in HSR.

**Correction notice** This article has been corrected since it first published online. The open access licence type has been amended.

**Acknowledgements** We thank the Dutch HSR institutions that participated in this study: Erasmus MC, Department of Public Health; Erasmus University, Erasmus School of Health Policy and Management; Leiden University Medical Centre, Department of Medical Decision Making and the Department of Public Health and Primary Care; University Maastricht, Health Services Research group; the Netherlands Institute for Health Services Research (NIVEL); Radboud UMC, IQ healthcare; the National Institute for Public Health and the Environment (RIVM); University of Groningen, Faculty of Economics and Business; Tilburg University, Social and Behavioural Sciences, Tranzo; Trimbos Institute; University Medical Centre Utrecht, Julius Centre for Health Sciences and Primary Care; Amsterdam UMC – Vrije Universiteit Amsterdam, and the Amsterdam UMC - University of Amsterdam. Furthermore, we also would like to thank project advisers Anton Kunst, Thomas Plochg and Patrick Romano. We thank Ellen Nolte, Enrique Bernal-Delgado, Michel Wensing, Oliver Groene and Wilm Quentin for their valuable feedback on the definition of QRPs and the measurement instrument.

**Contributors** DSK, NSK, MJB, RGG and TJ designed the study. RGG, TJ and JM collected the data. RGG and JM analysed the data, and RGG drafted the manuscript. TJ, JM, MJB, NSK and DSK were all involved in the interpretation of the results and were major contributors to the writing of the manuscript. All authors read and approved the final manuscript and are accountable for all aspects of the work.

**Funding** This study was funded by ZonMw grant number 445001003.

**Disclaimer** The funder had no role in the study design, in the collection, analysis, and interpretation of data, in the writing of the manuscript, and in the decision to submit the paper for publication. All authors had full access to the data during the conduct of this study and take responsibility for the integrity of the data and the data analysis.

**Competing interests** None declared.

**Patient consent for publication** Not required.

**Ethics approval** Medical ethics review committee at Amsterdam UMC.

**Provenance and peer review** Not commissioned; externally peer reviewed.

**Data sharing statement** The data are available upon request to the corresponding author.

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
