## [Reviewer comments · BMJ Open]

ARTICLE DETAILS

TITLE (PROVISIONAL)	The occurrence and nature of questionable research practices in the reporting of messages and conclusions in international scientific Health Services Research publications: A structured assessment of publications authored by researchers in the Netherlands
AUTHORS	Gerrits, Reinie; Jansen, Tessa; Mulyanto, Joko; van den Berg, Michael; Klazinga, Niek; Kringos, Dionne

VERSION 1 - REVIEW

REVIEWER	Joeri Tijdkink Amsterdam UMC, location VU medical Center, The Netherlands
REVIEW RETURNED	26-Nov-2018

GENERAL COMMENTS	General comments: The study describes innovative research and gives a new perspective on reporting of research results in a specific domain (HSR). This significantly adds knowledge to the discussion on spin and selective messages and reporting. The methods are sound and the results are well written. It has the potential to spark the discussion on responsible reporting of research results without overselling or overstating them. They have adequately described their methods and have used a thorough analysis of the problem. However, I do feel that some room for improvement in reporting of methods, results and conclusion. The three most important concerns is the use of the term QRPs, the measurement instrument and the lack of severity/a taxonomy of the QRPs. First, QRPs are generally associated with poor research practices and although we do not consider it fraud, in most surveys on QRPs, we define QRPs as bad behavior. In this study the authors may want to rethink the use of the word QRP for this. Although I do understand that all the practices described are questionable, it might be more useful and adequate if they use another term (ie irresponsible reporting of results? Irregular interpretation of conclusions? (probably the authors have better terms/definitions). It may be less confusing if they use another term or definition in the title and abstract. Secondly, the items in the measurement instrument leave room for improvement. The authors have used several judgments to determine the existence of the QRP in the study. However, if I review the items, I have doubts that most items do not fully and precisely describe whether it is questionable or not and why the observers/authors have made decisions to decide whether the
--

item is a QRP. These items leave room for discussion and evoke problems with the interpretation of the items. (IE what is poorly documented exactly? What it adequately reflect? What is exactly when a research question is differently phrased in the introduction and the discussion? What is unjustifiable use of hyperboles? In the appendix they try to resolve this issue by further explain the items. I feel that this still leaves room for debate and give enough degrees of freedom to have different opinions about what should be considered a QRP and what not. This is also an argument that the authors may want to change the term QRPs.

Finally, the construction of 35 items is very interesting and thorough and this gives the reader enough food for thought. However, these 35 items differ extremely from one another so I can image that a construction of a taxonomy is a very helpful way to guide readers through the items and to give substantial help to adequately interpret the results. This will help to decide which items are more concerning, severe and more important than others. Did the authors think of this option to construct a taxonomy or a severity scale (ie with the use of the interviews they have conducted during the construction of the instrument)? I think it may be helpful for readers to better understand and interpret the results.

Furthermore, below some minor remarks on the different headings

Introduction

Great introduction. The only thing I was wondering is why the field of HSR is so important to study. As you may know, in all other biomedical fields studies spin their results. Why studying HSR publications is different in this? Are there grounds to believe that HSR is intrinsically different than other fields?

Secondly, the authors hardly refer to earlier research. IS there a specific reason not to reflect on earlier evidence? What about reflecting on the surveys that are conducted on QRPs. Most surveys are self reported but may add and spark the discussion as a starting point and another ground to decide that this manuscript is more rigorous and has no self reporting bias.

Methods:

The validation of the measurement instrument is described. However, I think it can use some more information to better understand the validation techniques. How was the consensus meeting organized? How did you manage that all institutions agreed on the methods? Was there a research protocol? What the 14 interviews with directors/leaders looked like. Did you use a semi structured interview protocol and is it possible to publish this protocol? How is it validated exactly? Is this validation described in another paper or preprint? Is there an analysis plan or preregistration of the validation study and the survey?

Finally, it is not completely clear how the selection of publications is organized. I do understand why they chose not to publish the list of publications for ethical reasons. However, it might be helpful to give an example how the selection of the publications took place. Page 5: 'All data were entered in the DEF'. I might have missed this but it would be nice to link it to the actual data evaluation form in the manuscript/appendix

Results:

	No specific comments, other than that some items are multi-interpretible. This results in an average of 5-6 QRPs per publication and this is extremely high, also according to the meta-analysis of Fanelli in 2011. Is it fruitful to discuss this difference in the discussion section? Discussion: The authors elaborate on the strengths and weaknesses of the study. They describe the degree of subjectivity as a weakness but think that their consensus method may fail as I am not fully convinced that all QRPs are defined in detail. It may be helpful to compare their findings with other papers on spin and QRPs in the past. (ie Fanelli's work PLOS One 2009, many other surveys on spin) The authors ask in the end of the discussion to urge editors to join the debate. How can they further steer this discussion? Are there more recommendations they can make, based on their data?
--	---

REVIEWER	Lauren Maggio Uniformed Services University, United States
REVIEW RETURNED	07-Dec-2018

GENERAL COMMENTS	Thank you for this opportunity to review this manuscript on questionable research practices. I believe this is an important topic that will be of interest to BMJ Open readers. Below are several comments, which I hope the authors will consider to strengthen their manuscript. Introduction Please provide additional description of questionable research practices preferably early in the introduction. Ideally, you would include a general definition of QRPs. I realize that in the method section you provide the consensus generated definition for HSR, but this feels too late and is specific to that field. The introduction starts out quite broad in relation to QRPs. However, based on the method this study appears to be focused on QRPs related to the reporting of messages and conclusions. This focus excludes some forms of QRPs, such as those related to self-citation, salami slicing, etc., that a reader might expect to be explored. Therefore, it would be helpful to explicitly state this focus in the introduction. Convert "What is already known" and "Added value of this study" into narrative format and integrate this information into the existing paragraphs. Additionally, references are needed to support the included claims. A reference would be especially important to support the claim that 100% of publications containing rhetorical practices resulting in spin. Method The paragraph on page 4 starting on line 48 may benefit from a flow diagram to depict the inclusion/exclusion decisions. Thank you for including the data extraction form and the detailed instructions.
---

	The acronym DEF is distracting. Please spell this out. This sentence is unclear: As a result, two identified QRPs were retracted, and two QRPs were added. Does this mean that two QRPs were retracted / added to the DEF or that these were modified for the 6 publications that were reassessed? It is noted that NK and DK reassessed a random sample of 6 publications, so 10%. Did they each review 6 papers each for a total of 12? This is unclear. -Please provide an example of a QRP that was judged to not be assessable in the case of qualitative research (page 5). Additionally, were there any QRPs that were specific to qualitative research? Was this considered in the creation of the QRP listing? Results / Discussion I would be interested to know more about the journals in which the included papers were published, beyond the journal impact factor. For example, how many journals were represented in the sample? From my perspective, journals can have an impact on how authors present their research and report key elements. For example, an author preparing a manuscript for a journal that has a 2700 word count vs. 3500 word count might approach their reporting differently. Other factors that may play a role could be the author guidelines for particular journals, whether or not they require authors to adhere to specific reporting guidelines, etc. The role of the journal may be an additional topic to consider for the discussion.
--	---

REVIEWER	Professor Andrea Jorgensen University of Liverpool, UK
REVIEW RETURNED	17-Jan-2019

GENERAL COMMENTS	 1. I feel that the phrase 'Questionable Research Practice' needs a clearer definition in the introduction section so that the reader has an understanding from the outset of what it means. Alternatively, the authors need to make it clearer from the outset that the idea of 'Questionable Research Practice' was developed and defined as part of the study via consensus meeting and literature review. 2. Further terminology that requires more explanation is 'rhetorical practices resulting in spin'. 3. Has the literature review and results of the consensus meeting been published elsewhere ? If not, then it is recommended that the authors provide further details of this procedure e.g. what was the search strategy for the literature review, details of the search results, what was the format of the consensus meeting, summary of discussions at the meeting etc. This could be a stand alone paper or an appendix to the current paper. The same goes for the process of developing the final list of QRPs - details of this process are too sparse and it would be valuable to understand how it was developed - particularly for those interested in using the list in future. 4. I find description of the sampling approach rather confusing. Why did papers have to have a unique first author? What is justification for this ? Further, why was number from each institution capped at 10 ? What is justification for the sample size ?
--

	5. Please clarify how agreement between the 4 reviewers was assessed and what is the 'second round' referred to when discussing agreement ? 6. In the section on Analysis, what do the authors mean by 'subject to the scale' ? 7. The involvement mentioned under the section on 'Patient and Public involvement' does not really relate to the involvement of patients or public and I suggest it is removed, or at least changed to explain no such involvement occurred (assuming that to be the case). 8. In table 1, why do numbers in each Study Design category not equal the sample size, in total ? 9. In figure 2 it would be useful to provide counts for each study also (i.e. number of QRPs in each study), for completeness. 10. I feel that the data in Table 3 should also be presented graphically given its importance to the paper's findings.
--	---

VERSION 1 – AUTHOR RESPONSE

Reviewer(s)' Comments to Author:

Reviewer: 1

Reviewer Name: Joeri Tjink

Institution and Country: Amsterdam UMC, location VU medical Center, The Netherlands Please state any competing interests or state 'None declared': none declared

Please leave your comments for the authors below General comments:

The study describes innovative research and gives a new perspective on reporting of research results in a specific domain (HSR). This significantly adds knowledge to the discussion on spin and selective messages and reporting. The methods are sound and the results are well written. It has the potential to spark the discussion on responsible reporting of research results without overselling or overstating them. They have adequately described their methods and have used a thorough analysis of the problem.

However, I do feel that some room for improvement in reporting of methods, results and conclusion. The three most important concerns is the use of the term QRPs, the measurement instrument and the lack of severity/a taxonomy of the QRPs.

First, QRPs are generally associated with poor research practices and although we do not consider it fraud, in most surveys on QRPs, we define QRPs as bad behavior. In this study the authors may want to rethink the use of the word QRP for this. Although I do understand that all the practices described are questionable, it might be more useful and adequate if they use another term (ie irresponsible reporting of results? Irregular interpretation of conclusions? (probably the authors have better terms/definitions). It may be less confusing if they use another term or definition in the title and abstract.

Reply: we thank the reviewer for his interest in our paper, and his careful consideration of our applied terminology and methodology. The term questionable has raised some discussion amongst our stakeholder groups as well, and we have discussed the use of the term for this paper extensively.

We agree with the reviewer that some of the included QRPs in this paper are less severe, while others might be considered unacceptable. The term 'questionable' raises concerns about severity, however, we want to emphasize that in essence the term means that a topic 'raises questions'. As

other terms we have considered, such as 'irresponsible', ascribe more blame to the author than might be fair, or 'inadequate', which provides a more normative stance on the practices being wrongful. The term 'questionable' leaves room for discussion on both the severity of the QRP, whether the QRP is acceptable, and where the responsibility for addressing the QRP lies. By identifying a questionable research practice in this study, we indicate: this reporting of messages and conclusions raises questions. Moreover, 'questionable research practices' is as a term commonly used in current research debates on fostering responsible research practices in the Netherlands, including the research program funded by the Netherlands Organisation for Health Research and Development (ZonMw). In line with this terminology, through this study, the field of Health Services Research joins in on this debate. To address the reviewers' feedback, we further we clarified this position at the start of the introduction, and added our definition to the methods section in the abstract.

Revision: We have elaborated on our operationalization in the introduction section on line 76-81 and in the methods section of the abstract.

Secondly, the items in the measurement instrument leave room for improvement. The authors have used several judgments to determine the existence of the QRP in the study. However, if I review the items, I have doubts that most items do not fully and precisely describe whether it is questionable or not and why the observers/authors have made decisions to decide whether the item is a QRP. These items leave room for discussion and evoke problems with the interpretation of the items. (IE what is poorly documented exactly? What it adequately reflect? What is exactly when a research question is differently phrased in the introduction and the discussion? What is unjustifiable use of hyperboles? In the appendix they try to resolve this issue by further explain the items. I feel that this still leaves room for debate and give enough degrees of freedom to have different opinions about what should be considered a QRP and what not. This is also an argument that the authors may want to change the term QRPs.

Reply: As in any review of the quality of scientific reporting, there is subjectivity in the judgement of a QRP. Throughout the assessment process we were consistent in our decision making, applying a systematic process for identifying each QRP. In its core, these decisions were based on the following questions: was a statement made? Was some justification, argumentation for the statement made following or preceding it? Does the statement follow logically from this argument? Two reviewers first did this individually, then together during the consensus meeting. In this process we had few disagreements in the interpretation of the logic. In two cases the reviewers were uncertain of their mutual judgement and asked for advice. In the supplementary material we provide the Data Extraction Form containing a description of the considerations made for each QRP.

Revisions: to provide more insight in the decisions made to identify a QRP and the judgement used by the reviewers, we have added some explanation to the assessment strategy and basis for decision of the reviewers in the section 'assessment process' on line 192-194.

Finally, the construction of 35 items is very interesting and thorough and this gives the reader enough food for thought. However, these 35 items differ extremely from one another so I can image that a construction of a taxonomy is a very helpful way to guide readers through the items and to give substantial help to adequately interpret the results. This will help to decide which items are more concerning, severe and more important than others. Did the authors think of this option to construct a taxonomy or a severity scale (ie with the use of the interviews they have conducted during the construction of the instrument)? I think it may be helpful for readers to better understand and interpret the results.

Reply: We agree that the development of a taxonomy would be very useful. However, since this study is one of the first in the area of HSR, the scope was explorative and a rating of severity per QRP was beyond its scope. We have had some internal discussions and such a taxonomy seems to be not straightforward, and interviewees differ in their opinion about this. While one HSR leader may

expressed the absence of recommendations as absolutely unacceptable in HSR, another interviewee did not think of it as a severe problem. These differences in opinion on the contents or strictness of an HSR publication ask for closer investigation in the normative stance of HSR researchers at different levels before determining such a taxonomy. Consequently, we recommend the HSR community to debate the severity of these QRPs and determine norms in the reporting of scientific literature in the field of HSR.

Revisions: no revisions made.

Reviewer 1: Furthermore, below some minor remarks on the different headings	Reply
Introduction Great introduction. The only thing I was wondering is why the field of HSR is so important to study. As you may know, in all other biomedical fields studies spin their results. Why studying HSR publications is different in this? Are there grounds to believe that HSR is intrinsically different than other fields?	Reply: This combination of HSR specific characteristics may result in a different set of QRPs in the reporting of a scientific study. The broad section of study designs may lead to an increase in unjustified claims of causality, the context specific research may increase unjustified claims of generalisability, and the difficulty in translating knowledge to practice may result in unsupported recommendations or implications. Revision: we elaborated on the importance of HSR in the introduction on line 99-103.
Secondly, the authors hardly refer to earlier research. IS there a specific reason not to reflect on earlier evidence? What about reflecting on the surveys that are conducted on QRPs. Most surveys are self reported but may add and spark the discussion as a starting point and another ground to decide that this manuscript is more rigorous and has no self reporting bias.	Reply: the literature that we used as background was primarily summarized in a systematic review by Chiu et al. and a literature description by Boutron et al. (2018) who describe the research on 'spin' in biomedical studies. We appreciate the advice of the reviewer to include Survey studies to compare our methodology and results to. Revision: In the discussion we have added a comparison to the review of Fanelli (2010) to the limitations section on line 311-315.
Methods: The validation of the measurement instrument is described. However, I think it can use some more information to better understand the validation techniques. How was the consensus meeting organized? How did you manage that all institutions agreed on the methods? Was there a research protocol? What the 14 interviews with directors/leaders looked like. Did you use a semi structured interview protocol and is it possible to publish this protocol? How is it validated exactly? Is this validation described in another paper or preprint? Is there an analysis plan or preregistration of the validation study and the survey?	Reply: The construction of the checklist was not published elsewhere. Revision: In line with the reviewers' comments, we have added an extended description of the construction of the definition and measurement instrument to the supplementary material.

Finally, it is not completely clear how the selection of publications is organized. I do understand why they chose not to publish the list of publications for ethical reasons. However, it might be helpful to give an example how the selection of the publications took place.

Revision: In the section 'sample' on line 157-162, we added some information regarding the selection process.

Page 5: 'All data were entered in the DEF'. I might have missed this but it would be nice to link it to the actual data evaluation form in the manuscript/appendix.

Reply: The Data Extraction Form is included in the supplementary material for the use and consideration of the readers.

Results:
No specific comments, other than that some items are multi-interpretable. This results in an average of 5-6 QRPs per publication and this is extremely high, also according to the meta-analysis of Fanelli in 2011. Is it fruitful to discuss this difference in the discussion section?

Reply: After this review we have considered the meta-analysis of Fanelli and compared it to our results. Fanelli's review focusses on (self-reported) misconduct and questionable research practices in science. Rather than the less severe, more likely unconscious questionable research practices that we assessed, Fanelli focusses on the knowledge of conscious, and perhaps more established QRPs. Considering the difference in severity of the items that we reviewed, it is difficult to compare the outcomes of this research. Comparing this number to the instances of 'spin' in biomedical literature (see a review by Chiu et al. (2017) (which varies between 10% and 100% depending on the definition used), the occurrence of QRPs in HSR is not remarkably high in our opinion.

Discussion:
The authors elaborate on the strengths and weaknesses of the study. They describe the degree of subjectivity as a weakness but think that their consensus method may fail as I am not fully convinced that all QRPs are defined in detail.

Reply: as stated previously, we have included the data extraction form with extended explanations of the included items.

It may be helpful to compare their findings with other papers on spin and QRPs in the past. (ie Fanelli's work PLOS One 2009, many other surveys on spin) The authors ask in the end of the discussion to urge editors to join the debate. How can they further steer this discussion? Are there more recommendations they can make, based on their data?

Reply: we have included the publication by Fanelli et al. (2010). The QRPs they identify are different to ours. Therefore, it is difficult to make a direct comparison between the results. We believe it is necessary to first set norms and identify the factors that would prevent these QRPs from occurring. After norms and expectations are set, there is a role for HSR institutions as well as journals in implementing policies and practices that support them. We have added this norm direction in the last line of the section 'Implications and recommendations for policy and practice'.

Reviewer: 2

Reviewer Name: Lauren Maggio

Institution and Country: Uniformed Services University, United States Please state any competing interests or state 'None declared': None declared

Thank you for this opportunity to review this manuscript on questionable research practices. I believe this is an important topic that will be of interest to BMJ Open readers. Below are several comments, which I hope the authors will consider to strengthen their manuscript.

Reviewer 2	Reply
Introduction Please provide additional description of questionable research practices preferably early in the introduction. Ideally, you would include a general definition of QRPs. I realize that in the method section you provide the consensus generated definition for HSR, but this feels too late and is specific to that field. The introduction starts out quite broad in relation to QRPs. However, based on the method this study appears to be focused on QRPs related to the reporting of messages and conclusions. This focus excludes some forms of QRPs, such as those related to self-citation, salami slicing, etc., that a reader might expect to be explored. Therefore, it would be helpful to explicitly state this focus in the introduction.	Reply: we thank the reviewer for their valuable comments to improve our manuscript. We agree with the necessity for clarity on the definition of the term 'questionable', and have added a section early in the introduction explaining our approach to the term 'questionable research practices' and refer to our specified definition later on. Revision: we have elaborated on our operationalisation of QRPs in the introduction on line 76-81.
Convert "What is already known" and "Added value of this study" into narrative format and integrate this information into the existing paragraphs. Additionally, references are needed to support the included claims. A reference would be especially important to support the claim that 100% of publications containing rhetorical practices resulting in spin.	Revision: we have included the section 'what is already know' in the text on line 85-90, explained the rhetorical practices, and added the reference for this claim (Chiu et al. 2017). The content of the section 'added value of this study' was already was already integrated in the introduction of the manuscript.
Method The paragraph on page 4 starting on line 48 may benefit from a flow diagram to depict the inclusion/exclusion decisions.	Reply: we agree with the reviewer that visualisation of complex selection processes generally improves clarity, however only three steps were taken in our selection process: selection by HSR title, randomized selection of publications, and the replacement of two publications. During our writing process we made such a diagram, but as it only contains 3 boxes, we believe it would not increase understanding of the selection process.
Thank you for including the data extraction form and the detailed instructions.	Revision: we have removed the acronym.

The acronym DEF is distracting. Please spell this out.

This sentence is unclear: As a result, two identified QRPs were retracted, and two QRPs were added. Does this mean that two QRPs were retracted / added to the DEF or that these were modified for the 6 publications that were reassessed?

It is noted that NK and DK reassessed a random sample of 6 publications, so 10%. Did they each review 6 papers each for a total of 12? This is unclear.

-Please provide an example of a QRP that was judged to not be assessable in the case of qualitative research (page 5). Additionally, were there any QRPs that were specific to qualitative research? Was this considered in the creation of the QRP listing?

Results / Discussion

I would be interested to know more about the journals in which the included papers were published, beyond the journal impact factor. For example, how many journals were represented in the sample? From my perspective, journals can have an impact on how authors present their research and report key elements. For example, an author preparing a manuscript for a journal that has a 2700 word count vs. 3500 word count might approach their reporting differently. Other factors that may play a role could be the author guidelines for particular journals, whether or not they require authors to adhere to specific reporting guidelines, etc. The role of the journal may be an additional topic to consider for the discussion.

Reply: in total 12 publications were reassessed (10% of the total selection). The QRPs were modified for the reassessed publications.

Revision: we have added this information to the section 'assessment process' on line 191-193.

Reply: an example of a QRP that was not applicable to qualitative research was e.g. effect size is overestimated. We added this example in section 'analysis' on line 201-202. No items that are only applicable to qualitative research were included, all the items that would be applicable to qualitative research were also applicable to quantitative research. While we did specifically address the inclusion of qualitative research, we found no unique QRPs for qualitative research in the existing checklists or in the interviews.

Reply: 80 different journals were included in the sample. We agree with the reviewer that the journal restrictions and regulations may have a strong impact on the published manuscript. This includes word restrictions, journal guidelines, and the approach to the review process.

Revision: We added the number of journals to table 1. We have added to the discussion to focus journal and institutional factors on line 351-353. Additionally, we added the influencing factor of the researching institution.

Reviewer: 3

Reviewer Name: Professor Andrea Jorgensen Institution and Country: University of Liverpool, UK
Please state any competing interests or state 'None declared': None declared

Reviewer 3	Reply
I feel that the phrase 'Questionable Research Practice' needs a clearer definition in the introduction section so that the reader has an understanding from the outset of what it means. Alternatively, the authors need to make it clearer from the outset that the idea of 'Questionable Research Practice' was developed and defined as part of the study via consensus meeting and literature review.	Reply: In line with the comments made by the previous reviewers, we agree that we need to further clarify the term 'questionable'. Revision: We added a section in the introduction explaining how we applied the term in our research on line 76-81.
Further terminology that requires more explanation is 'rhetorical practices resulting in spin'.	Revision: We added an explanation of the meaning of 'rhetorical practices resulting in spin' in the introduction on line 85-90.
Has the literature review and results of the consensus meeting been published elsewhere ? If not, then it is recommended that the authors provide further details of this procedure e.g. what was the search strategy for the literature review, details of the search results, what was the format of the consensus meeting, summary of discussions at the meeting etc. This could be a stand alone paper or an appendix to the current paper. The same goes for the process of developing the final list of QRPs - details of this process are too sparse and it would be valuable to understand how it was developed - particularly for those interested in using the list in future.	Reply: The construction of the checklist was not published elsewhere. Revision: following the comments by the reviewers, we have added an extended description of the construction of the measurement instrument to the supplementary material.
I find description of the sampling approach rather confusing. Why did papers have to have an unique first author? What is justification for this ? Further, why was number from each institution capped at 10 ? What is justification for the sample size ?	Reply: the papers have a unique first author as the primary author may be expected to have most influence on what is written. We intended to get an overview of the occurrence of QRPs in the Netherlands, including publications with the same authors would have biased the sample; instead a wide variation of authorship is desirable. The limitation of the sample size and the cap of 10 is created to include an equal number of publications of institutes, as institute culture and policy is another factor that may influence the occurrence of QRPs. It was capped at 10 (and not higher) for feasibility reasons. Revisions: We have added this explanation to the section 'sample' on line 147-150.

Please clarify how agreement between the 4 reviewers was assessed and what is the 'second round' referred to when discussing agreement ?	Reply: we added some clarification on the test process and the consensus process in the manuscripts on line 88-190.
In the section on Analysis, what do the authors mean by 'subject to the scale' ?	Reply: we intended to indicate that we adjusted the mean to the appropriate number (e.g. mean impact factor was calculated over 93 publications) Revision: we agree with the reviewer that the term is confusing and have removed it from the manuscript.
The involvement mentioned under the section on 'Patient and Public involvement' does not really relate to the involvement of patients or public and I suggest it is removed, or at least changed to explain no such involvement occurred (assuming that to be the case).	Reply: There was no involvement of patient and the public. This heading however is required the journal. Nevertheless, the involvement of the institutions (as representatives of our study population) is vital to our study and we would like to report on their involvement. Revision: To improve transparency, we have added the sentence, 'No patients were involved in this study.'
In table 1, why do numbers in each Study Design category not equal the sample size, in total?	Reply: the description of the study designs 'meta-analysis' and 'case study' was missing from this table in the manuscript. Revision: We added the study designs to table 1.
In figure 2 it would be useful to provide counts for each study also (i.e. number of QRPs in each study), for completeness.	Reply: we agree this would provide additional insight in the spread of the QRPs across publications. Revisions: we have added the count of QRPs to figure 2.
I feel that the data in Table 3 should also be presented graphically given its importance to the paper's findings.	Reply: we agree that the findings in this table are an important part of this paper. Figure 2 visualises the number of QRPs across publications and provides insight in these findings as well. While the figure does not show percentages (like in table 3), it does show the count of QRPs, what QRPs are most prevalent, and their relation to the other QRPs, in addition to displaying the dispersion of QRPs across our sample. With the addition of QRP count to this figure, we believe visualizing the findings in table 3 separately would not provide new insights to the readers of the paper.

VERSION 2 – REVIEW

REVIEWER	Joeri Tijdink, MD PhD Amsterdam UMC, location VUmc Department of medical humanities
REVIEW RETURNED	14-Feb-2019

GENERAL COMMENTS	Great work. I find it very elegant that you have addressed most issues raised in the review in the strengths and limitation sections..
--

REVIEWER	Professor Andrea Jorgensen University of Liverpool, UK
REVIEW RETURNED	14-Feb-2019

GENERAL COMMENTS	Thank you very much for addressing my previous comments, I feel that you have added clarity to some parts which were previously unclear. Congratulations on a very well written manuscript. I just have one remaining comment - I don't find figure 2 particularly clear. In particular, if the studies are ordered from least to most QPRs, should they not be aligned vertically ? Can you please clarify ? Thank you.
--

VERSION 2 – AUTHOR RESPONSE

Reviewer(s)' Comments to Author:

Reviewer: 1

Reviewer Name: Joeri Tijdink, MD PhD

Institution and Country: Amsterdam UMC, location VUmc
Department of medical humanities

Please state any competing interests or state 'None declared': none declared

Please leave your comments for the authors below Great work. I find it very elegant that you have addressed most issues raised in the review in the strengths and limitation sections..

Reply: We thank the reviewer for his compliments and the work put into improving our manuscript.

Reviewer: 3

Reviewer Name: Professor Andrea Jorgensen

Institution and Country: University of Liverpool, UK

Please state any competing interests or state 'None declared': None declared

Please leave your comments for the authors below Thank you very much for addressing my previous comments, I feel that you have added clarity to some parts which were previously unclear.

Congratulations on a very well written manuscript. I just have one remaining comment - I don't find figure 2 particularly clear. In particular, if the studies are ordered from least to most QRPs, should they not be aligned vertically ? Can you please clarify ? Thank you.

Reply: We thank the reviewer for her compliments, and we are happy to answer her last question. The aim of the graph is to display the occurrence of QRPs across the sample, and how QRPs may occur together in a single publication. If we would align the QRPs vertically, we would lose visual information on how many QRPs occur. With the addition of "the number of QRPs" above the graph during the last revision, we believe we have added the necessary information on the number of QRPs occurring across the sample.